

# Trajectory patterns and factors influencing perinatal fatigue among Chinese women from late pregnancy to 6 months after delivery

Xiaoxiao Zhu and Haiou Xia

School of Nursing, Fudan University, Shanghai, China

## ABSTRACT

**Background:** Perinatal fatigue among women is related to the clinical outcomes of mothers and infants. Perinatal fatigue changes over time, and the trajectory varies according to the predictors and outcomes of the mothers. This, however, has not been documented in any study.

**Objectives:** This study aimed to identify and characterize the trajectory patterns of perinatal fatigue among women from late pregnancy to 6 months after delivery.

**Methods:** We used growth mixture modeling to estimate the trajectory of perinatal fatigue at 28 gestational weeks (T0), 37 gestational weeks (T1), 3 days (T2), 1 week (T3), 6 weeks (T4), and 6 months (T5) after delivery with ($n = 1,030$). The Mann-Whitney U test and binary logistic regression were used to tie the selected trajectory classes to predictors and outcomes.

**Results:** There were two distinct patterns of perinatal fatigue in women: "persistently high" (11.1%, $n = 114$) and "persistently low" (88.9%, $n = 916$). Levels of perinatal fatigue among women in the "persistently high" group were higher than those in the "persistently low" group across the six measurements. Complications, fatigue at T0, and employment status in late pregnancy were all significant predictors of trajectories. Additionally, the "persistently high" group had a greater prevalence of difficult baby care and weight retention and a lower prevalence of exclusive breastfeeding.

**Conclusions:** Our study proved the heterogeneity and characteristics of perinatal fatigue among women. Future research should concentrate on developing intervention packages targeted at specific individuals in order to alleviate perinatal fatigue in women.

Corresponding author
Haiou Xia, hsxia@fudan.edu.cn

## INTRODUCTION

Perinatal fatigue, which includes both pregnancy and postpartum fatigue, is one of the most frequently reported complaints among childbearing women (*Cheng et al., 2015*; *Liu et al., 2020*). Perinatal fatigue is related to adverse clinical outcomes for women and infants. Prenatal fatigue could increase rates of pregnant morbidity (*Luke et al., 1999*), cesarean
section (*Chien & Ko, 2004*), preterm birth, and even postpartum depression (*Newman et al., 2001*). Postpartum fatigue may delay the initiation of breastfeeding (*Saidi & Godbout, 2016*), reduce mother-infant attachments (*Lai et al., 2015*), child development (*Dokuhaki, Dokuhaki & Akbarzadeh, 2021*) and even cause postpartum depression (*Wilson, Lee & Bei, 2019*). Collectively, comprehensive understandings of perinatal fatigue among women are needed.

However, longitudinal data on perinatal fatigue in women are quite scarce. Longitudinal studies have mainly focused on the levels and courses of perinatal fatigue among the population over time, which may obscure individual differences (*Cheng & Pickler, 2014*; *Cheng et al., 2015*; *Iwata et al., 2018*). Only one study, Kuo's investigation, has reported perinatal fatigue trajectories among 121 women from the late pregnancy to 1 week postpartum (*Kuo et al., 2012*). Kuo's analysis clarified the heterogeneity of fatigue among women from pregnancy to postpartum periods. However, class 3 of the perinatal fatigue trajectories in Kuo's study only had nine women due to its limited sample sizes, and it only measured perinatal fatigue among women until 1 week postpartum, which shortened perinatal periods. Thus, it is necessary to gain understanding into whether and how perinatal fatigue trajectories emerge in larger samples of women throughout longer perinatal periods.

Investigating the factors that contribute to prenatal fatigue trajectories is also critical for intervention development. Few aspects of perinatal fatigue trajectories have been reported, though multiple factors are associated with the perinatal fatigue (*Badr & Zauszniewski, 2017*; *Yehia et al., 2020*). In this study, the theoretical framework for factors influencing perinatal fatigue was according to Pugh's framework (*Pugh & Milligan, 1993*). Pugh's framework used the Classification of Nursing Diagnosis definition of fatigue: "An overwhelming sustained sense of exhaustion and decreased capacity for physical and mental work." In this framework, related factors of perinatal fatigue were categorized into situational, physiologic, psychologic, and performance factors (Fig. 1). Investigating the factors that influence perinatal fatigue trajectories may provide to a better understanding of how to avoid or reduce perinatal fatigue in women.

From late pregnancy to 6 months after birth, this prospective, longitudinal study assessed perinatal fatigue in women six times. The particular objectives of this study were to detect and define the trajectory patterns of perinatal fatigue in women from late pregnancy to 6 months after birth. Extending our understanding of the identified perinatal fatigue trajectory types and their related components may help to improve clinical management for perinatal fatigue.

## MATERIALS AND METHODS

### Study design and settings

This study is a prospective, longitudinal study, conducted between December 2020 and September 2021. Participants were asked to fill out questionnaires six times, namely, 28 gestational weeks ($T_0$), 37 gestational weeks ($T_1$), 3 days postpartum ($T_2$), 1 week postpartum ($T_3$), 6 weeks postpartum ($T_4$), and 6 months postpartum ($T_5$). This work was

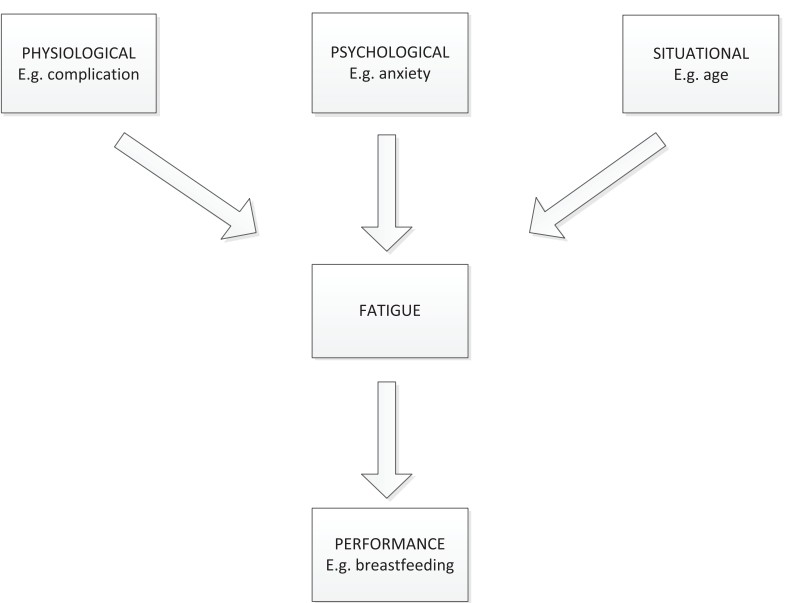

**Figure 1 A framework for the study of mother's fatigue during the childbearing experience.**

approved by the Obstetrics and Gynecology Hospital attributed to Fudan University (No. 2020-191).

## Participants

We used the continuous enrollment method to find eligible participants between December 2020 and January 2021. Pregnant woman at (28 + 0, 29 + 6) weeks was scanned and recruited. The exclusion criteria were women with (a) diagnosed depression before; (b) severe health problems such as heart failure; (c) twins or multiples. All women signed informed consent.

## Measurements

Perinatal fatigue was measured by the Postnatal Accumulated Fatigue Scale (PAFS). Professor Emi Mori firstly published the scale in 2016 (*Tsuchiya et al., 2016*). Each of the 13 items of this scale has a three-point rating from 0–3 (0 = rarely, 1 = sometimes and 3 = often). These 13 items are divided into three dimensions (physical, emotional, and cognitive), for example, "I feel unfocused" (cognitive). The total score ranges from 0 to 39, with higher scores indicating higher fatigue levels. This scale has good reliability and validity in the former study (*Tsuchiya et al., 2016*). In this study, we obtained authorization from Professor Emi Mori regarding the use of the PAFS, and translated the Japanese version of PAFS into a Chinese version. The Chinese version of PAFS scale also has good reliability (Cronbach's alpha = 0.823–0.908) and validity (KMO = 0.801–0.91) in this sample. Data of perinatal fatigue were collected six times from $T_0$ to $T_5$.

Self-made questionnaires collected data of situational factors at $T_0$ and $T_2$. At $T_0$, we collected data of maternal age, parity (primipara, multipara), employment (employed, unemployed), an education level (<college, college, >college), monthly family income

(<¥10,000, ¥10,000–20,000, >¥20,000) and residency (urban, suburban). Also, we collected data of planned pregnancy (yes, no), prenatal training (yes, no), sleep duration per day (<6, 6 to 8, >8 h), and sleep quality at this time. Of them, sleep quality was evaluated by asking women, "Please score your sleep quality in the past 24 h, ranging from 0 to 10, and 0 score means very dissatisfied, and 10 scores mean very satisfied." Besides, fetus gender (boys, girls) was obtained by medical records at $T_2$.

Self-made questionnaires were also used to collect data of physiologic factors at $T_0$ and $T_1$. Maternal height, prenatal weight, weight at $T_0$, gestation, complication, pain, and fatigue were collected at $T_0$. Of them, the pain was measured by asking one question: "Please score your pain in the past week, ranging from 0 to 10, and 0 score means no pain and 10 scores mean very painful." Based on these data, we calculated prenatal body mass index (BMI), BMI at $T_0$, weight gain at $T_0$ (weight at $T_0$ minus prenatal weight), and BMI gain at $T_0$ (BMI at $T_0$ minus prenatal BMI). And "$T_1 - T_0$ fatigue" was equal to PAFS fatigue score at T1 minus that at T0, similar calculations were used for $(T_2 - T_1)/(T_3 - T_2)/(T_4 - T_3)/(T_5 - T_4)$ Fatigue.

One maternal psychologic factor, anxiety, was measured at $T_0$ by asking participants: "How anxious are you in the past week? Choose a number between 0–10 to express the extent of your anxiety, 0 for no anxiety, 1 to 3 scores for mild anxiety, 4 to 6 scores for moderate anxiety, 7 to 9 scores for severe anxiety, and 10 scores for extreme anxiety."

Performance factors among women were regarded as outcomes of perinatal fatigue in this study, which were all collected at $T_5$, including difficult baby care (yes, no), exclusive breastfeeding (EBF; yes, no), and weight (kilogram, M ± SD). Also, weight retention was defined as "yes" if weight at $T_5$ was more than prenatal weight, *vice versa* as "no."

## Statistical analysis

Only data of participants who completed at least three follow-ups were included in the analysis. Data analysis was carried out in the following two parts based on the two objectives of this study.

In the first part, we used growth mixture modeling (GMM) to identify the trajectory of perinatal fatigue. GMM is an advanced, person-centered analysis to identify heterogeneous subgroups that comprise distinct courses across time (*Feldman, Masyn & Conger, 2009*). The time intervals were uneven from $T_0$ to $T_5$, namely, 0 weeks $(T_0 - T_0)$, 8 weeks $(T_1 - T_0)$, 12 weeks $(T_2 - T_0)$, 13 weeks $(T_3 - T_0)$, 18 weeks $(T_4 - T_0)$, 36 weeks $(T_5 - T_0)$, respectively. Thus, the time score of $T_0$ to $T_5$ was fixed here at 0, 8, 12, 13, 18, 36, representing the equidistant time model. A robust maximum likelihood estimator was used for model estimation. Akaike Information Criterion (AIC), Bayesian Information Criterion (BIC), and Sample-Size Adjusted BIC (aBIC) were used as the model fit goodness statistics, with smaller values suggesting better model fit. Entropy was used to evaluate the accuracy of classification. The closer the entropy was to 1, the higher the class accuracy was. Likelihood Ratio Test (LRT) compared an n-class model with an n-1-class model, where a significant *P* value indicated a considerable improvement in the n-class model and thus rejected n-1-model rationally. In this study, the Lo-Mendell-Rubin (LMR) and Bootstrap Likelihood Ratio Test (BLRT) were also used for model comparison. Bayesian

posterior probability was applied to export latent trajectory classes after a best-fit model was determined. In this part, GMM was performed with Mplus 8.3. In addition, a rank-sum test was used to examine differences in fatigue scores at six times and during five adjacent times between the trajectories. And $P < 0.05$ were preliminary remarked as positive.

In the second part, to characterize the trajectory of perinatal fatigue, the Mann-Whitney U test, and binary logistic regression were implemented. We calculated the Z-score of skewness and kurtosis for all continuous variables (Table S1). Only gestation at baseline was normally distributed, but it does not have homogeneity of variance. Thus, the Mann-Whitney U test was used to compare differences between all variables, including continuous variables (such as maternal age) and categorical variables (such as monthly family income). Variables with $P < 0.05$ were preliminary remarked as positive between the trajectory classes. To further describe characteristics of perinatal fatigue trajectory, binary logistic regression (Forward: LR) was applied. The latent trajectory classes as the dependent variable and positive potential predictors as independent variables in the regression model. And variables with $P < 0.05$ in the regression model were regarded as predictors of perinatal fatigue trajectory. In this part, performance factors were not included in the regression model because they were considered outcomes rather than potential predictors. SPSS version 23.0 was used for Mann-Whitney U and binary logistic regression.

# RESULTS

Of 1,150 women who met initial eligibility criteria and consented to participation, a total of 1,030 women who completed at least three follow-ups were included in this report. Cases included in each visit were 1,009 ($T_0$), 917 ($T_1$), 970 ($T_2$), 781 ($T_3$), 563 ($T_4$), and 779 ($T_5$), respectively (Fig. 2). The average age was 30.47 (SD 4.01) years in this study. Besides, 75.8% of these women were primiparas, 82.1% had a college degree, 80.9% were employed, and 87.1% had high monthly family income(≥¥10,000). No significant difference of Basic information was found between included and non-included subjects (Table S2).

## Identifying perinatal fatigue trajectories

Table 1 shows the GMM of one to five fatigue trajectory classes, including 1-class model, 2-class model, 3-class model, 4-class model, and 5-class model. All models can be supposed to obtain the best maximum likelihood. The best fit model is based on different model fit indices: (1) the AIC, BIC, and ABIC are smaller; (2) the entropy is closer to 1; (3) LMR and BLRT are both significant. Collectively, the 2-class model was the best fit model in this study.

The mean intercept factor of Class 1 (12.318) was higher than that of Class 2 (10.118), and the variance of the intercept factor between these two classes was 13.586 ($P < 0.001$), indicating a relatively higher initial fatigue score in Class 1 than Class 2. The average slope factors of Class 1 and Class 2 were 0.306 ($P < 0.001$) and −0.110 ($P < 0.001$), indicating that Class 1's fatigue score increased significantly over time while Class 2's decreased significantly over time. In addition, the trajectories of fatigue scores were also

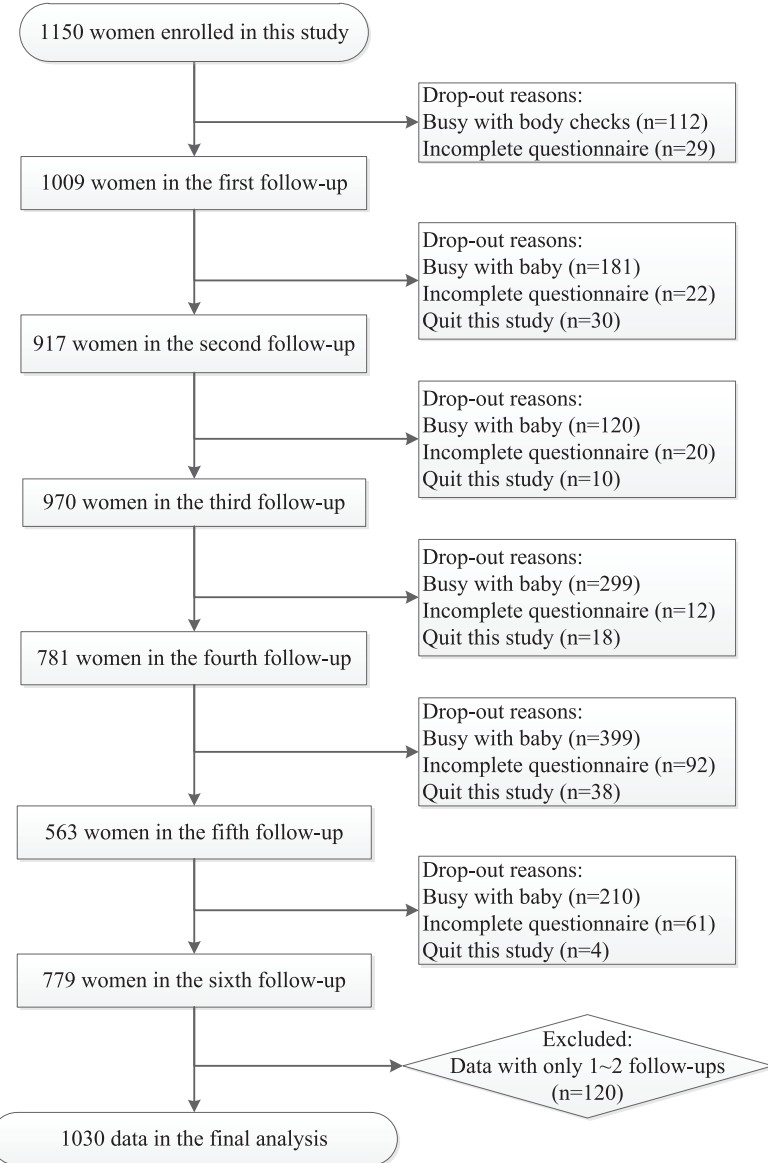

**Figure 2  A flowchart of the sample election and follow-ups from late pregnancy to 6 months after delivery in this study.**               

visually shown in Fig. 3. Thus, Class 1 and Class 2 were named as the "persistently high" group (11.1%; $n = 114$) and "persistently low" group (88.90%; $n = 916$), respectively.

Table 2 showed that fatigue scores had inconsistent growth tendency between different adjacent times. Firstly, an interesting finding was that the fatigue score in the "persistently high" group increased significantly faster than the "persistently low" group from $T_0$ to $T_1$ ($P = 0.001$). And no significant differences in fatigue scores were found from $T_1$ to $T_2$, from $T_2$ to $T_3$, and from $T_3$ to $T_4$. Combined Table 1 with Fig. 1, from $T_0$ to $T_4$, fatigue scores in both classes presented similar upward (from $T_0$ to $T_3$) and downward (from $T_3$ to $T_4$) trends. Oppositely, fatigue scores showed a strong upward trend from $T_4$ to $T_5$ in the "persistently high" group, whereas a solid downward trend in the "persistently low"

**Table 1 Fit indices for one to five fatigue trajectory classes of GMM.**

| C | k | AIC | BIC | aBIC | E | P of LMR | P of BLRT | Class probability |
|---|---|-----|-----|------|---|----------|-----------|-------------------|
| 1 | 11 | 33,257.239 | 33,311.549 | 33,276.612 | – | – | – | 1 |
| 2 | 14 | 33,022.681 | 33,091.803 | 33,047.338 | 0.865 | 0.0000 | 0.0000 | 0.908/0.092 |
| 3 | 17 | 32,989.331 | 33,073.266 | 33,019.272 | 0.821 | 0.4590 | 0.0000 | 0.850/0.096/0.053 |
| 4 | 20 | 32,964.370 | 33,063.117 | 32,999.594 | 0.831 | 0.0504 | 0.0000 | 0.830/0.101/0.057/0.012 |
| 5 | 23 | 32,951.673 | 33,065.231 | 32,992.181 | 0.798 | 0.3609 | 0.0000 | 0.772/0.109/0.097/0.012/0.011 |

Note:

C, number of classes; k, number of free parameters; AIC, Akaike Information Criterion; BIC, Bayesian Information Criterion; aBIC, sample size adjusted BIC; E, Entropy; P of LMR, $p$-value of Lo-Mendell-Rubin test; P of BLRT, $p$-value of Bootstrap Likelihood Ratio Test; class probability, proportion of sample classification.

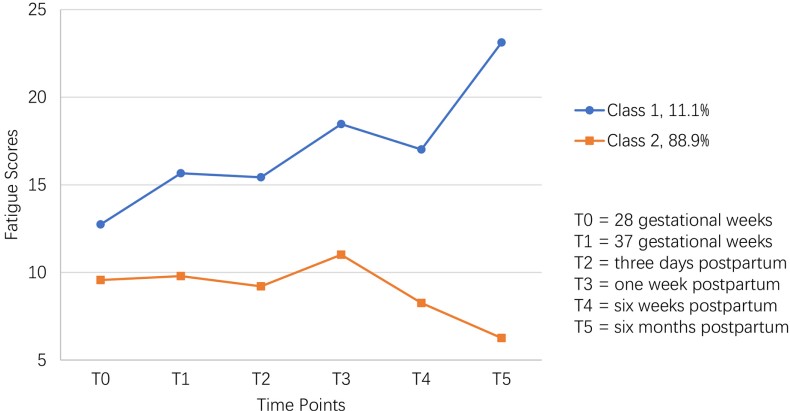

Figure 3 legend:
- Class 1, 11.1%
- Class 2, 88.9%

T0 = 28 gestational weeks
T1 = 37 gestational weeks
T2 = three days postpartum
T3 = one week postpartum
T4 = six weeks postpartum
T5 = six months postpartum

**Figure 3 The two-class model of perinatal fatigue trajectory among women from late pregnancy to 6 months after delivery in this study.**

group ($P < 0.001$). In addition, for the "persistently high" group, the peak fatigue score showed at $T_5$, followed by that at $T_3$. For the "persistently low" group, the peak of fatigue score showed at $T_3$.

## Characteristics of perinatal fatigue trajectories

Table 3 summarizes the two groups' descriptive statistics of related factors (situational, physiologic, psychologic, and performance factors). For situational factors, individuals in the "persistently high" group were less employed ($P = 0.002$) and had worse sleep quality ($P < 0.001$) at $T_0$ than those in the "persistently low" group. For physiologic factors, women in the "persistently high" group had smaller prenatal weight ($P = 0.014$), weight at $T_0$ ($P = 0.014$), BMI at $T_0$ ($P = 0.036$), more complication ($P < 0.001$), and severer fatigue at $T_0$ ($P < 0.001$) than those in "persistently low" group. For psychologic factors, more people in the "persistently high" group reported anxiety status with scores between 7–10 at $T_0$ than the "persistently low" group did ($P < 0.001$).

Binary logistic regression was used to analyze the relationships between potential factors and the perinatal fatigue trajectory. The regression model was summarized in Table 4. In the model, perinatal fatigue trajectory classes were the dependent variable and coded as 1 = "persistently high" group and 0 = "persistently low" group. And all factors significantly

**Table 2 Fatigue scores at six times and during five adjacent times.**

| Variables | Persistently high (m, SD, M) | Persistently low (m, SD, M) | U | Z | P |
|---|---|---|---|---|---|
| Fatigue at $T_0$ | 12.74(6.312), 12.00 | 9.57(5.455), 9.00 | 55,314.500 | 4.937 | <0.001 |
| Fatigue at $T_1$ | 15.66(9.53), 15.50 | 9.79(7.373), 8.00 | 46,577.500 | 5.399 | <0.001 |
| Fatigue at $T_2$ | 15.43(7.894), 14.00 | 9.2(6.813), 7.00 | 59,067.000 | 7.509 | <0.001 |
| Fatigue at $T_3$ | 18.47(9.336), 18.00 | 11.01(7.483), 10.00 | 38,128.500 | 6.703 | <0.001 |
| Fatigue at $T_4$ | 17.02(10.846), 18.00 | 8.26(6.323), 6.00 | 23,391.500 | 6.071 | <0.001 |
| Fatigue at $T_5$ | 23.12(4.518), 22.00 | 6.26(3.993), 6.00 | 52,148.000 | 14.188 | <0.001 |
| $T_1$–$T_0$ Fatigue | 3.28(8.864), 2.00 | 0.38(7.602), 0.00 | 38,087.500 | 2.651 | 0.008 |
| $T_2$–$T_1$ Fatigue | 0.05(11.23), −1.00 | −0.73(7.713), 0.00 | 33,258.000 | 0.131 | 0.896 |
| $T_3$–$T_2$ Fatigue | 2.86(11.28), 3.00 | 1.62(8.839), 1.00 | 23,955.000 | 0.778 | 0.436 |
| $T_4$–$T_3$ Fatigue | −0.83(14.432), 0.00 | −3.06(8.791), −4.00 | 16,427.500 | 0.456 | 0.648 |
| $T_5$–$T_4$ Fatigue | 9.62(10.012), 11.00 | −2.1(6.504), −1.00 | 12,552.500 | 6.947 | <0.001 |

Note:
Mann-Whitney U test. m, mean; SD, standard deviation; M, median; $T_0$, 28 gestational weeks; $T_1$, 37 gestational weeks; $T_2$, 3 days postpartum; $T_3$, 1 week postpartum; $T_4$, 6 weeks postpartum; $T_5$, 6 months postpartum. $T_1$ − $T_0$ Fatigue = fatigue at $T_1$ minus that at $T_0$. $(T_2 − T_1)/(T_3 − T_2)/(T_4 − T_3)/(T_5 − T_4)$ Fatigue were named in similar way.

**Table 3 Descriptive statistics of factors between the trajectory classes.**

| Variables | Persistently high | Persistently low | U | Z | P |
|---|---|---|---|---|---|
| Number of participant, $n$(%) | 95(9.2) | 935(90.8) | – | – | – |
| Situational factors | | | | | |
| Age, m(SD), M, y | 30.21(3.57), 30 | 30.45(3.66), 30 | 40,745.5 | −0.959 | 0.337 |
| Parity, $n$(%) | | | 42,199.0 | −1.013 | 0.311 |
| Primipara | 76(80.0) | 702(75.1) | | | |
| Multipara | 19(20.0) | 230(24.6) | | | |
| Missing | 0(0.0) | 3(0.3) | | | |
| Employment, $n$(%) | | | 46,473.0 | 3.049 | 0.002 |
| Employed | 62(65.3) | 749(80.1) | | | |
| Unemployed | 28(29.5) | 163(17.4) | | | |
| Missing | 5(5.2) | 23(2.5) | | | |
| Education level, $n$(%) | | | 44,252.0 | 0.506 | 0.613 |
| <college | 7(7.4) | 73(7.8) | | | |
| college | 16(16.8) | 175(18.7) | | | |
| >college | 71(74.7) | 671(71.8) | | | |
| Missing | 1(1.1) | 16(1.7) | | | |
| Monthly family income, $n$(%) | | | 40,010.5 | −0.489 | 0.625 |
| <¥10,000 | 12(12.6) | 118(12.6) | | | |
| ¥10,000–20,000 | 47(49.5) | 455(48.7) | | | |
| >¥20,000 | 31(32.6) | 342(36.6) | | | |
| Missing | 5(5.3) | 20(2.1) | | | |
| Residency, $n$(%) | | | 41,431.5 | −1.824 | 0.068 |
| Urban | 93(97.9) | 891(95.3) | | | |
| Suburban | 0(0) | 32(3.4) | | | |
| Missing | 2(2.1) | 12(1.3) | | | |

| Variables | Persistently high | Persistently low | U | Z | P |
|---|---|---|---|---|---|
| Planned pregnancy, $n$(%) | | | 39,788.5 | −1.543 | 0.123 |
| Yes | 59(62.1) | 646(69.1) | | | |
| No | 35(36.8) | 271(29.0) | | | |
| Missing | 1(1.1) | 18(1.9) | | | |
| Prenatal training, $n$(%) | | | 39,422.0 | −1.451 | 0.147 |
| Yes | 17(17.9) | 233(24.9) | | | |
| No | 75(78.9) | 687(73.5) | | | |
| Missing | 3(3.2) | 15(1.6) | | | |
| Sleep duration, $n$(%), hours | | | 44,526.0 | 1.898 | 0.058 |
| <Six | 9(9.5) | 97(10.4) | | | |
| Six to eight | 53(55.8) | 651(69.6) | | | |
| >Eight | 26(27.4) | 173(18.5) | | | |
| Missing | 7(7.4) | 14(1.5) | | | |
| Sleep quality, m(SD), M | 7.00(1.10), 7 | 7.48(1.19), 8 | 31,530.5 | −4.252 | <0.001 |
| Fetus gender, $n$(%) | | | 37,380.5 | 1.222 | 0.222 |
| Boys | 39(41.1) | 425(45.5) | | | |
| Girls | 47(49.5) | 388(41.5) | | | |
| Missing | 9(9.5) | 122(13.0) | | | |
| Physiologic factors | | | | | |
| Height, m(SD), M, cm | 161.99(4.79),1.62 | 162.76(4.93),1.63 | 39,954.5 | −1.332 | 0.183 |
| Prenatal weight, m(SD), M | 53.71(7.97),52.00 | 55.78(8.24),55.00 | 35,223.0 | −2.456 | 0.014 |
| Prenatal BMI, m(SD), M | 20.44(2.64),20.19 | 21.04(2.83),20.70 | 36,474.5 | −1.967 | 0.050 |
| Weight at $T_0$, m(SD), M | 62.89(8.22),61.00 | 65.20(8.66),65.00 | 35,899.0 | −2.487 | 0.013 |
| BMI at $T_0$, m(SD), M | 23.93(2.66), 23.63 | 24.59(2.96), 24.43 | 36,905.0 | −2.094 | 0.036 |
| Weight gain at $T_0$, m(SD), M | 9.05(2.83), 9.00 | 9.43(3.48), 9.05 | 38,264.5 | −1.132 | 0.258 |
| BMI gain at $T_0$, m(SD), M | 3.45(1.06), 3.61 | 3.56(1.31), 3.54 | 38,957.0 | −0.849 | 0.396 |
| Gestation, m(SD), M | 29.07(0.70), 29.14 | 29.18(0.74), 29.14 | 40,055.5 | −0.882 | 0.378 |
| Complication, $n$(%) | | | 28,058.5 | −3.877 | <0.001 |
| YES | 27(28.4) | 139(14.9) | | | |
| NO | 53(55.8) | 710(75.9) | | | |
| Missing | 15(15.8) | 86(9.2) | | | |
| Pain, m(SD), M | 2.54(2.70), 1.50 | 2.03(2.28), 2.00 | 36,871.0 | 1.264 | 0.206 |
| Fatigue at $T_0$, m(SD), M | 12.74(6.312), 12.00 | 9.57(5.455),9.00 | 55,314.5 | 4.937 | <0.001 |
| Psychologic factor | | | | | |
| Anxiety, $n$(%) | | | 46,235.0 | 5.419 | <0.001 |
| One to three | 12(12.6) | 243(26.0) | | | |
| Four to six | 12(12.6) | 230(24.6) | | | |
| Seven to nine | 17(17.9) | 194(20.7) | | | |
| Ten | 41(43.2) | 168(18.0) | | | |
| Missing | 13(13.7) | 100(10.7) | | | |

(Continued)
| Table 3 (continued) | | | | | |
|---|---|---|---|---|---|
| Variables | Persistently high | Persistently low | U | Z | P |
| Maternal outcomes | | | | | |
| Difficult baby care, $n$(%) | | | 32,522.0 | 4.706 | <0.001 |
| YES | 35(36.8) | 158(16.9) | | | |
| NO | 39(41.1) | 546(58.4) | | | |
| Missing | 21(22.1) | 231(24.7) | | | |
| Exclusive breastfeeding, $n$(%) | | | 20,217.5 | −3.102 | 0.002 |
| YES | 31(32.6) | 439(47.0) | | | |
| NO | 40(42.1) | 263(28.1) | | | |
| Missing | 24(25.3) | 233(24.9) | | | |
| Weight at $T_5$, m(SD), M | 57.48(7.95), 56.50 | 56.98(8.34), 55.00 | 26,815.5 | 0.844 | 0.399 |
| Weight retention at $T_5$, $n$(%) | | | 18,802.5 | −3.281 | 0.001 |
| YES | 55(57.9) | 380(42.0) | | | |
| NO | 15(15.8) | 293(30.0) | | | |
| Missing | 25(26.3) | 262(28.0) | | | |

**Note:**
Mann-Whitney U test. m, mean; SD, standard deviation; M, median; $n$, number; %, percentage; y, year(s); cm, centimeters(s); h, hour(s); BMI, body mass index; $T_0$, 28 gestational weeks; $T_5$, 6 months postpartum.

**Table 4 Binary logistic regression model testing predictors of fatigue trajectory classes.**

| Variable | B | SE | W | df | P | OR | 95% CI OR |
|---|---|---|---|---|---|---|---|
| Complication | 1.022 | 0.269 | 14.453 | 1 | <0.001 | 2.779 | [1.641–4.706] |
| Fatigue at $T_0$ | 0.083 | 0.021 | 15.553 | 1 | <0.001 | 1.086 | [1.042–1.132] |
| Employment | −0.609 | 0.287 | 4.508 | 1 | 0.034 | 0.544 | [0.310–0.954] |
| constant | −3.101 | 0.356 | 75.992 | 1 | <0.001 | 0.045 | – |

**Note:**
Binary Logistic Regression. B, regression coefficient; SE, standard error; W, ward; df, degree of freedom; P, $p$-value; OR, odds ratio; CI, confidence interval.

differed between the two classes were regarded as independent variables. But a high correlation was found between fatigue at $T_0$ and sleep quality (r = 0.38, $P < 0.001$), fatigue at $T_0$ and anxiety (r = 0.36, $P < 0.001$), BMI at $T_0$ and prenatal weight (r = 0.79, $P < 0.001$), and BMI at $T_0$ and Weight at $T_0$ (r = 0.87, $P < 0.001$). Thus, only employment (1 = "employed" and 0 = "unemployed"), complication (1 = "yes" and 0 = "no"), fatigue at $T_0$, and BMI at $T_0$ from 878 cases were introduced into the final model. The χ2 test of the final model was significant ($P < 0.001$), and the Hosmer and Lemeshow test indicated an acceptable model fit with $P = 0.417$. As a result, the final model showed three predictors of perinatal fatigue trajectories, *i.e.*, complication (odds ratio [OR], 2.779, 95% CI [1.641–4.706]), fatigue at T0 (OR, 1.086, 95% CI [1.042–1.132]) and employment (OR, 0.544; 95% CI [0.310–0.954]).

In addition, Table 3 also showed differences in the performance factors between the two groups. The percentages of patients with difficult baby care (36.8% *vs* 16.9%; $P < 0.001$) and weight retention (56.8% *vs* 40.9%; $P = 0.001$) were significantly higher in the

"persistently high" group than the other group at $T_5$. Also, fewer people reported exclusive breastfeeding (EBF, 32.6% *vs* 47%; *P* = 0.002) in the "persistently high" group than the other group at $T_5$.

## DISCUSSIONS

Our study addressed previously unresolved gaps in the literature by identifying and describing the perinatal fatigue trajectory in a large sample from late pregnancy to 6 months postpartum. The findings of this study are critical for developing therapies to alleviate perinatal fatigue and improve mother and child clinical outcomes. We discovered two unique perinatal fatigue trajectories, namely "persistently high" (class 1, 11.1%) and "persistently low" (class 2, 88.9%). Additionally, we were the first to identify predicted markers for the perinatal fatigue trajectory, such as complication, fatigue at $T_0$, and employment. Additionally, women in the "persistently high" group experienced more adverse clinical outcomes than those in the "persistently low" group, including greater rates of difficult baby care and weight retention and lower rates of exclusive breastfeeding at $T_5$.

The identification of two distinct trajectories of prenatal fatigue in women confirmed the variability of perinatal fatigue. Though few research have shown this in pregnant women, prior investigations have identified two distinct fatigue trajectories in other populations, such as patients with cancer (*Bødtcher et al., 2015*) and systemic sclerosis (*Willems et al., 2017*). To our knowledge, only Kuo's study has investigated this in perinatal women, identifying three distinct trajectories (class 1, 44.6%; class 2, 48.0%; and class 3, 3.7%) of perinatal fatigue from late pregnancy to 1 week postpartum in 121 women (*Kuo et al., 2012*). Kuo's study detected one additional trajectory of perinatal fatigue in women than ours, which could be explained by the difference in measurement frequency, follow-up time, and sample size. Further research could be undertaken to confirm the trajectories of perinatal fatigue in women. Additionally, both our study and Kuo's study discovered that around one-tenth (11.1% *vs* 7.4%) of women were at high risk of experiencing extreme fatigue during their perinatal periods. Clinical personnel should identify and monitor these high-risk women more closely.

Notably, levels of perinatal fatigue were significantly higher in women classified as "persistently high" than in women classified as "persistently low" at all six time points from late pregnancy to 6 months after delivery, which was consistent with prior study in other patients (*Willems et al., 2017*; *Li et al., 2020*; *Bean et al., 2021*). Additionally, the general level of prenatal fatigue grew dramatically with time in the "persistently high" group, whereas it decreased significantly in the "persistently low" group. This characteristic revealed that fatigue could be closely related to specific internal components.

Two time points have been identified for decreasing perinatal fatigue in women. One was 1 week postpartum for women classified as "persistently low." This conclusion is consistent with previous research emphasizing the critical role of 1 week postpartum in preventing perinatal fatigue (*Henderson, Alderdice & Redshaw, 2019*; *Fata & Atan, 2018*). The other was 6 months after birth for women classified as having "persistently high" blood pressure. On the other hand, Taylor's study found that perinatal fatigue reached a

minimum 6 months following delivery (*Taylor & Johnson, 2010*). In this study, the lowest level of perinatal fatigue was also 6 months after delivery in the "persistently low" group. This discrepancy may be explained by Taylor's study's analysis of the peak of perinatal fatigue using trends rather than trajectories. Thus, for the majority of women, the peak of perinatal weariness occurred 1 week after delivery, whereas for others, it occurred 6 months later. Future research could examine the factors that contribute to fatigue in women 1 week and 6 months following delivery.

In our investigation, a complication was identified as a risk factor for developing an elevated perinatal fatigue pattern. Women who experienced complications during late pregnancy, such as GDM (*Zhang et al., 2017*) or hypertension (*Li et al., 2016*), reported higher level of perinatal fatigue. Complications have also been implicated as risk factors for fatigue in pregnant women (*Mortazavi & Borzoee, 2019*) and postpartum women (*Taylor & Johnson, 2013*). Pathways for preventing or alleviating problems during late pregnancy should be researched further.

Another intriguing discovery was that exhaustion during late pregnancy may have an effect on how subsequent perinatal fatigue develops over time. Women who had increased fatigue throughout late pregnancy were more likely to fall into the "persistently high" category. Although this discovery has not been replicated in other studies, a similar pattern has been observed in prenatal depression (*Nège et al., 2020*). Thus, health care providers could identify women at risk of extreme exhaustion throughout the perinatal period as early as late pregnancy. Additionally, future research could develop predictive models to validate the prediction of perinatal fatigue in women based on exhaustion during late pregnancy.

Employment status was also associated with perinatal fatigue trajectories. This result suggested that women whose employment ratio was "persistently high" had a lower employment ratio during late pregnancy. Similarly to our employment finding, Chien's study found that unemployed women experienced considerably higher levels of weariness than other women during pregnancy (*Chien & Ko, 2004*). This difference could be explained by job insecurity, which has been shown to impair psychological resilience (*Heinz et al., 2018*). As a result, women should be encouraged to seek employment rather than evade it in late pregnancy.

Additionally, to examine determinants of perinatal fatigue trajectories in women, we evaluated variations in performance parameters such as challenging baby care ratios, EBF, and weight retention at 6 months postpartum across the two trajectories. We discovered that the "persistently high" group had a larger proportion of challenging baby care than the other group, which is consistent with previous reports (*Lai et al., 2015*; *Henderson, Alderdice & Redshaw, 2019*). Both of these investigations, however, were cross-sectional. Thus, our study was the first to reveal the detrimental effect of perinatal fatigue on 6-month-old newborn care.

Does fatigue impact breastfeeding in women during their reproductive years? This issue has dogged health care providers for 15 years (*Saidi & Godbout, 2016*; *Callahan, Sejourne & Denis, 2006*). At 6 months postpartum, our study discovered that the ratio of exclusive breastfeeding was considerably lower in the "persistently high" group than in the

"persistently low" group. Despite the cross-sectional nature of the study, earlier research has also demonstrated a detrimental relationship between postpartum fatigue and breastfeeding outcomes (*Brown et al., 2014*; *Aldalili & El Mahalli, 2021*). Logically, a higher level of postpartum fatigue among women was related to delayed breastfeeding initiation (*Heinig, 2010*), decreased self-efficacy (*Fata & Atan, 2018*), and increased breastfeeding difficulties (*Cooklin et al., 2018*), all of which finally result in the termination of exclusive breastfeeding (*Gianni et al., 2019*). Our study was the first to establish a link between the prenatal fatigue trajectory and the outcomes of exclusive breastfeeding 6 months after delivery, using a different study design and data analysis method than previous studies.

In addition, women need to escape weight retention (*Durst et al., 2015*; *Martin et al., 2015*; *Sharkey et al., 2016*). At 6 months postpartum, 56.8 percent of women in the "persistently high" group experienced weight retention, compared to 40.9 percent of women in the "persistently low" group. To our knowledge, no other study has examined the association between perinatal fatigue and weight retention in postpartum women. Additional research is required to determine the association between fatigue and weight retention.

Several limitations should be highlighted regarding this investigation. To begin, our study stopped collecting data on perinatal fatigue 6 months after birth. Fatigue may occur in women who deliver more than a year after delivery (*Parks et al., 1999*; *Wilson et al., 2018*). A longer duration is necessary for examining the trajectories of perinatal fatigue in women. Second, the exact instrument used to assess perinatal fatigue was developed in 2016 and is not yet widely available. There is a need for reports on the implementation of the PAFS scale in perinatal mothers from diverse cultural backgrounds. Third, anxiety was the sole psychological element during the perinatal period, according to Pugh's theoretical model of perinatal fatigue. As a result, we focused only on anxiety as a psychological component of perinatal fatigue. It was quantified using a single question rather than a predefined scale. Other psychologic variables, such as depression, should also be considered in future studies. Fourth, due to the researchers' limited time, only 574 women completed the 6-week postpartum follow-up. Additionally, 92.5% of our participants were college or university graduates. Our findings should be reproduced in lower-education populations.

This study has a few implications for perinatal care research and practice. For research purposes, future empirical investigations could validate and change the perinatal fatigue trajectory classes among women from many cultural backgrounds. Additionally, more predictive models should be developed to investigate and illustrate prenatal fatigue predictors. This will allow for future investigation of the characteristics of prenatal fatigue and the development of more effective therapies targeting sub-populations for lowering individual perinatal fatigue. Additionally, future research should examine the causal links between prenatal fatigue and clinical outcomes such as rates of difficult baby care, exclusive breastfeeding, and weight retention, thereby filling in the gaps in the data. In clinical practice, health care practitioners could undertake timely examinations of women early in their late pregnancy and screen out high-risk women who are at danger of experiencing extreme fatigue throughout the postnatal period.

To summarize, this study demonstrates the heterogeneity of perinatal fatigue in women by distinguishing two distinct perinatal fatigue trajectories: "persistently high" and "persistently low." Additionally, this study examined the characteristics of the two perinatal fatigue trajectories by finding three late-pregnancy predictors: complication, fatigue, and employment. Future research should focus on developing customized intervention packages based on these findings in order to alleviate women's perinatal fatigue.

## ACKNOWLEDGEMENTS

We express our great appreciation to all participants who provided data for this study and the experts who guided the design, ethical review, and implementation of this study.

### Funding

This work was supported by the National Natural Science Foundation of China (NSFC, No. 81701425) and the Nursing Scientific Research Fund of Fudan University (No. FNSF201903). The funders had no role in study design, data collection and analysis, decision to publish, or preparation of the manuscript.

### Grant Disclosures

The following grant information was disclosed by the authors:
National Natural Science Foundation of China: 81701425.
Nursing Scientific Research Fund of Fudan University: FNSF201903.

### Competing Interests

The authors declare that they have no competing interests.

### Author Contributions

- Xiaoxiao Zhu conceived and designed the experiments, performed the experiments, analyzed the data, prepared figures and/or tables, authored or reviewed drafts of the paper, and approved the final draft.
- Haiou Xia conceived and designed the experiments, performed the experiments, authored or reviewed drafts of the paper, and approved the final draft.

### Human Ethics

The following information was supplied relating to ethical approvals (*i.e.*, approving body and any reference numbers):

This work was approved by the Obstetrics and Gynecology Hospital attributed to Fudan University.

### Data Availability

The raw data are available in the Supplemental File.

## Supplemental Information

Supplemental information for this article can be found online at http://dx.doi.org/10.7717/peerj.13387#supplemental-information.

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
