# Peer review of "Trajectory patterns and factors influencing perinatal fatigue among Chinese women from late pregnancy to 6 months after delivery"

_PeerJ, doi:10.7717/peerj.13387_

## Round 0.1 · original submission · Major Revisions

Our reviewers have major concerns regarding the methodology quality of this paper. Please address these issues.

Reviewer 1 ·

Basic reporting

First, please check the English language of this paper and make necessary editing because I have found several problematic sentences and terms.
Second, the title seems not consistent with the work done by the authors because the authors classified subjects into two groups with different changing patterns of fatigue. The title means an overall changing patterns of fatigue, which should be provided in this paper.
Third, please consider whether logistic regression is appropriate to answer factors associated with the trajectory, I do not think these identified fixed factors can be associated a changing pattern.
Fourth, please consider latent variable growth model because there are many external indictors of the latent variable, fatigue.
Fifth, in the introduction, please clearly indicate the clinical significance of this research topic, i.e., whether the identified changing patterns could guide clinical intervention.
Sixth, the measure of fatigue is a foreign language version, I do not think it is valid for assessing fatigue in Chinese pregnant women.
Seventh, the psychological factor of anxiety is inadequate. Please provide your considerations, for example, why not included depression.
Eighth, the authors should provide a theoretical model of perinatal fatigue to inform the clinical research design of this study.

Experimental design

See above

Validity of the findings

See above

Additional comments

See above

Reviewer 2 ·

Basic reporting

Pugh’s framework were used as the theoretical framework to explore the related factors that influence perinatal fatigue. However, authors did not provide sufficient information to demonstrate how this framework work. Examples of the four types of factor - situational, physiologic, psychologic and performance - can be illustrated in the form of figure.

Experimental design

The methods and results sections of this paper leave out a lot of critical information, such as when the study was conducted, the flowchart of the participants which include information about drop-out reasons at each time point and a table comparing basic information about the included and non-included subjects. A checklist from the appropriate reporting guideline should be used to guide the author in writing.
Another issue is about the scale used to measure fatigue- Postnatal Accumulated Fatigue Scale (PAFS). The authors do not provide information on whether this scale has been translated and validated in the Chinese population.

Validity of the findings

one major problem with this article is that many of the conclusions in the discussion were not directly related to the results. For example,in line 235, the conclusion "fatigue may be strongly associated with individual internal substances. Interleukin-6 (IL-6) ..." seems very arbitrary.

Additional comments

no comment

---

## Round 0.2 · accepted · Accept

I am pleased to accept this revised paper.

Reviewer 1 ·

Basic reporting

None.

Experimental design

None.

Validity of the findings

None.

Additional comments

None.